# Comprehensive Identification of Deleterious *TP53* Missense VUS Variants Based on Their Impact on TP53 Structural Stability

**DOI:** 10.3390/ijms222111345

**Published:** 2021-10-20

**Authors:** Benjamin Tam, Siddharth Sinha, Zixin Qin, San Ming Wang

**Affiliations:** Cancer Centre and Institute of Translational Medicine, Faculty of Health Sciences, University of Macau, Taipa, Macau 999078, China; benjamintam@um.edu.mo (B.T.); siddharths@um.edu.mo (S.S.); yb87625@um.edu.mo (Z.Q.)

**Keywords:** TP53, VUS, deleterious, molecular dynamic simulations, ramachandran plot

## Abstract

TP53 plays critical roles in maintaining genome stability. Deleterious genetic variants damage the function of TP53, causing genome instability and increased cancer risk. Of the large quantity of genetic variants identified in TP53, however, many remain functionally unclassified as variants of unknown significance (VUS) due to the lack of evidence. This is reflected by the presence of 749 (42%) VUS of the 1785 germline variants collected in the ClinVar database. In this study, we addressed the deleteriousness of TP53 missense VUS. Utilizing the protein structure-based Ramachandran Plot-Molecular Dynamics Simulation (RPMDS) method that we developed, we measured the effects of missense VUS on TP53 structural stability. Of the 340 missense VUS tested, we observed deleterious evidence for 193 VUS, as reflected by the TP53 structural changes caused by the VUS-substituted residues. We compared the results from RPMDS with those from other in silico methods and observed higher specificity of RPMDS in classification of TP53 missense VUS than these methods. Data from our current study address a long-standing challenge in classifying the missense VUS in TP53, one of the most important tumor suppressor genes.

## 1. Introduction

As one of the most important and studied tumor-suppressor genes, TP53 plays essential roles in maintaining genome stability through controlling cell cycle, activating DNA damage repair, and initiating apoptosis [1,2,3]. *TP53* is also one of the most mutable tumor suppressor genes, affecting nearly all types of cancer [4]. While the majority of *TP53* variation is somatic, germline variation also frequently occurs and causes early development of multiple types of cancer, as represented by the Li-Fraumeni syndrome [5]. Efforts made in *TP53* germline variation studies have greatly enhanced our understanding of the mechanisms of tumorigenesis and have promoted the treatment, prognosis, and prevention of cancer.

Since *TP53* was identified four decades ago, 2177 *TP53* germline variants have been identified so far, including the 553 distinct *TP53* germline variants from the International Agency of Research on Cancer (IARC) *TP53* database (https://p53.iarc.fr, accessed on 6 June 2021) and the 1785 distinct *TP53* germline variants from the ClinVar database (https://www.ncbi.nlm.nih.gov/clinvar/, accessed on 18 June 2021). Unlike the frameshift or nonsense variants that commonly occur in many other tumor suppressors, germline variation in *TP53* is mostly missense variant, causing single amino acid substitution, and is located mostly within the DNA binding domain (DBD) of TP53. Although the pathogenicity of many hot-spot *TP53* germline variants, such as R175H, G245S, R248Q, R273C, R273H, and R282W [6], have been well determined, the function of the majority of *TP53* germline missense variants remains unknown due to the lack of functional evidence. Determination of the pathogenicity for the unclassified *TP53* missense variants remains an obstacle in translating the rich knowledge of TP53 in tumorigenesis into clinical cancer applications. This is also a common challenge in many cancer predisposition genes [7,8,9].

Protein structure has been widely used to study gene function [10,11] and has also been used to study the relationship between *TP53* variation and TP53 function [12]. We recently developed a protein-structure-based method, named Ramachandran Plot Molecular Dynamics Simulation (RPMDS), to study the effects of genetic variation on gene function [13]. In this method, the Ramachandran plot measures the influences of missense variants on TP53 secondary structure, and molecular dynamics simulation (MDS) simulates the dynamic effects of the variation on TP53 structural stability. By referring to the information from the known pathogenic, known benign/likely benign variants (herein referred as benign variants), and wild type alleles, RPMDS provides a quantitative measurement for the impact of missense variants on TP53 structural stability and uses the information to identify the deleterious missense variants. Testing of the RPMDS method in a group of *TP53* missense VUS showed satisfactory results [13].

In the current study, we performed a comprehensive analysis for all 340 *TP53* missense VUS present in the TP53 DNA binding domain by using the RPMDS method. We were able to provide physical evidence for the deleteriousness of 193 VUS based on their impact on TP53 structure.

## 2. Results

### 2.1. Construction of Mutant Protein Structures

From the ClinVar database, we identified a total of 443 missense variants located within the TP53 DNA binding domain, including 80 known pathogenic, 23 known benign/likely benign, and 340 VUS (Appendix A). Of the 443 variants, 146 (33%) were also recorded in the International Agency for Research on Cancer (IARC) TP53 database [14]. The TP53 DNA binding domain structure (PDB ID:2OCJ, 94–313 residues, 2.05 Å resolution) [15] was used as the template to construct TP53 mutant structures for each of the 443 missense variants. We compared the G245S-, R273C-, and R273H-based TP53 mutant structures with the experimentally determined TP53 crystal structures PDB ID:7DHY (97–289 residues), PDB ID: 4IBQ (87–288 residues), and PDB: 4IBS (96–288 residues), and we observed 99.48%, 99.48%, and 99.48% identity, respectively. The results show high reliability of the constructed mutant structures.

### 2.2. Identification of Deleterious Variants

We first used the data generated from the wild-type (WT) alleles, known pathogenic and benign variants, as the trained data to set the cut-off value for the deleterious variants, by following the process described in detail in our previous publication [13]. Briefly, the density deviation of the 80 known pathogenic variants was plotted against the normal and lognormal distribution curves for Anderson-Darling and Kolmogorov-Smirnov goodness-of-fit tests [16,17] (Figure 1A,B). Here, “structural deviation” was defined as the Ramachandran plot difference between the pathogenic and the trained data, of which the trained data contained the average benign variants and WT Ramachandran density plots. A significant deviation with a mean of 3.22 was obtained, with a scale sigma of 0.299 and upper and lower 95% boundaries between 3.158 and 3.491. The lower 95% boundary of the mean was set as the cut-off for the variants with deviation ≥ 3.158 to be classified as “deleterious”, and those <3.158 were classified as the “undefined” (no structure change doesn’t rule out deleterious effects). The deviation number was converted to a percentage to show the deviation rate from the WT. The Mann-Whitney test showed that the benign and pathogenic data were significantly different, with a Z score of 2.27 and *p*-value of 0.0223 [18] (Figure 1C).

Under the cut-off of 3.312, 47 of the 80 known pathogenic variants were classified as deleterious variants, with upper and lower 95% boundaries between 3.175 and 3.904 (23.9–49.6%) structural deviation from the trained data (Appendix A). The well-known pathogenic variants, including R175H, Y220C, G245S, R248Q, R273C, and R282W, had the highest structural deviation (39.3–49.6%) from the WT structure. The deleterious nature was well demonstrated by their destructive effects in the TP53 structure compared with the WT TP53 structure (Figure 2).

Under the deviation cut-off, 193 of the 340 missense VUS were classified as deleterious VUS, of which Y107D, M169V, R249S, T253N, and I255S had the highest structural deviation (>35%) (Appendix A).

### 2.3. Features of Deleterious VUS

We performed the following analysis for the classified 193 deleterious VUS variants:Deviation distribution. The range of deviation was 3.158–3.878 (23.5–48.3%) (Figure 3). This range was similar to that of known pathogenic variants, except for the top 3—R273C, Y220C, R175H (49.6, 47.4, 45.6%)—as they were highly destructive for the TP53 structure (Figure 2). For benign variants, 6 out of 23 benign variants were above the deleterious structure limits (>23.5%). Here, the Q165K had a structure deviation of 35.7% and was an outlier within the benign variants. The variants below the cutoff line were classified as “unknown”, as certain pathogenic variants may have minimal impact on structure stability [19]; therefore, their overall structure scaffolds can be comparable to benign variants. Overall, the results justified the use of the deviation from known pathogenic variants as the reference to classify the missense VUS.Spatial change of the substituted residues. The Ramachandran plot (RP) showed the spatial differences of the substituted residues from the wildtype residues, and the root-mean-square-deviation (RMSD) plot also showed the altered position of the substituted residues from the wildtype residues in the global TP53 structure (Figure 4A–C). For example, in S99F, F in RP showed its torsional angle in reflecting the rigidity of the fluctuation, and F in RMSD also showed its larger fluctuation, revealing its instability in the local environment; in G154R, R in RP showed its torsional angle fluctuation deviated from the wildtype G, and R in RMSD showed its large fluctuation, reflecting its instability in its local environment; in H214P, P in RP showed its torsional angle substantially fluctuated from the wildtype residue H, and the lower RMSD showed its high stability in TP53.Distribution in TP53 secondary structure. The deleterious VUS variants were distributed across the entire DNA binding domain, of which 44 were in the regions with few known pathogenic variants, including entire β sheets, loop 1 and loop 2, and all linkers; for the regions overlapped with the known pathogenic variants, the deleterious VUS variants were distributed more widely than the known pathogenic variants (Figure 5). The results indicate that RPMDS provides high sensitivity to detect deleterious missense VUS variants.Impact on TP53 local structure. The Ramachandran density plots showed that the deleterious VUS variants caused more local structural change, whereas known pathogenic variants caused more global structural change (Figure 6A). Taking Y107D, M169V, R249S, T253N, and I255S as examples (Figure 6B): in Y107D, the change of residue caused greater flexibility in the β strand (residues 108–114 (β1) and 204–208 (β6)); this was reflected by the diminishing peak in the Ramachandran density plot: an α helix loop was formed between residues 165–172, and this was not observed in WT; in M169V, the flexible structure was reflected by the diminished peaks at P-II and the β sheet region in the Ramachandran density plot. The extra α-helix bend was formed at residues 117–121 and 168–170, and α-helix 1 (177–181) was more structurally stable than the WT α-helix 1 (H1); with its high deviation of 3.770 (43.4%), R249S misfolded the TP53 structure globally, and the Ramachandran plot revealed a spike at the P-II region, while the β sheet region was diminished; in T253N, greater flexibility of β strand S4 and shortened β strand S8 (residues 156–161 and 230–236, respectively) reflected the diminished peak of the Ramachandran density plot; the residues 165–172 showed a stable structural formation, and this was not detected in the WT; in I255S, the Ramachandran density plot revealed the dissipated peak at the β sheet and P-II spiral regions; higher flexibility was present at the linker residues (244–246) and α-helix 2 (H2) (285–286), and S10 β strand (271–274) was extended by 2 residues.

### 2.4. Comparison with Other In Silico Methods

The ClinVar database periodically updates its classification for VUS variants based on new evidence. The 340 missense VUS variants used in our study were from the ClinVar database, accessed on 9 April 2020. By comparing the 340 missense VUS variants with the latest release of ClinVar VUS information, accessed on 17 May 2021, we identified 17 updated VUS classifications, of which 3 were re-classified as pathogenic and 14 as conflict interpretation (Table 1). The 3 VUS re-classified as pathogenic (c.695T>A, I232N; c.706TG, Y236D; c.794T>A, L265Q) were all classified as deleterious by our study; of the 14 VUS reclassified as conflict interpretation, 9 were classified as deleterious and 5 were classified as undefined by our study. The fact that RPMDS reached a similar classification for the 3 ClinVar re-classified VUS variants (I232N, Y236D, L265Q) demonstrates high specificity of RPMDS for deleterious variant classification. ClinVar re-classified 14 VUS as “conflict” implying that new deleterious evidence (although it may not be sufficient) had been accumulated for these VUS since their previous VUS classification. RPMDS provided structure-based evidence to support their deleterious effects.

We also compared the classification of the 340 missense VUS between RPMDS and 10 commonly used in silico methods, including Polyphen2_HDIV, Polyphen2_HVAR, SIFT, M-CAP, MutationTaster, LRT, PROVEAN, FATHMM, MetaSVM, and MetaLR (Table 1B, Appendix A). For the “deleterious” classification, RPMDS had the lowest rate, 56.8%, among all methods, of which FATHMM, MetaSVM, and MetaLR made 100% of deleterious classifications. The pattern was reversed for the “undefined” classification, where RPMDS was the highest, 43.2%, among all methods; FATHMM, MetaSVM, and MetaLR made none of “undefined” classifications. The results show that RPMDS provided the highest restriction in identifying the deleterious missense VUS as compared to other in silico methods.

## 3. Discussion

For the genetic variants identified, those causing substantial damaging consequences, such as large insertion/deletions that interrupt gene structure and the nonsense variants that create/mutate stop codon, can be easily identified as deleterious. However, it is difficult to judge the damaging impact for the missense variants, as they only cause single amino acid substitution, without obvious disruption of the overall protein structure. However, it has been well noticed that “common cancer mutants exhibit a variety of distinct local structural changes, while the overall structural scaffold is largely preserved” [19]. This is well reflected by the presence of abundant missense VUS variants in many cancer predisposition genes. Our analysis showed that, of the 92,563 variants from the 136 genes in the nine DNA damage repair pathways (ClinVar database, accessed 12 June 2021), 43,831 (47.4%) were VUS variants, of which 37,631 (85.9%) were missense VUS variants (Appendix A).

The challenge in missense VUS classification is the lack of evidence for its potential deleterious or benign impact. The presence of abundant missense VUS data suggests that it is impractical to fulfil the classification task by relying solely on experiment-based approaches, such as high-throughput genome editing methods, due to the cost restrictions, no matter how promising the approaches can be [20,21], or the classical bench-based methods due to their limited throughput capacity [22]. Instead, in silico approach has been highly regarded as a promising option to fulfil the task, due to its high-throughput capacity and low-cost nature. Continuous efforts have been made in this direction, as reflected by the many in silico methods developed thus far for variant classification, which have been designed by adapting different principles, including familial segregation [23], evolution conservation [24], statistics [25], computation [26], and experiment [27], or a combination of these principles [28,29,30,31,32,33,34,35,36,37,38]. While a decades’ application of these methods has demonstrated the power of each method in variant classification, their inherited limitation is also obvious, as described in the American College of Medical Genetics and Genomics and the Association for Molecular Pathology (ACMG/AMP) guidelines, which specified that “most tools also tend to have low specificity, resulting in over-prediction of missense changes as deleterious and are not as reliable at predicting missense variants with a milder effect” [39]. Multiple factors can contribute to the situation, including (1) different methods were designed using different principles. As such, a uniformed conclusion would not be expected, even for the same variants; (2) the “black box” effects for methods combining different principles. In contrast to expectations, it has been well determined that combination of different principles into one package does not necessarily function better than each individual method based on a single principle [40]. Testing new principles, such as machine-learning [41], is warranted to determine if they could improve the weaknesses of lower specificity and over-prediction of deleteriousness in the current in silico methods for variant classification.

Our design of RPMDS for missense VUS classification is based on the following considerations: (1) The system seeks evidence from only one phenotype, that is, the impact of variants on protein structure. This preference takes the advantage of existing well-determined protein structures for many cancer-related genes; (2) A missense VUS variant that is able to disturb the protein structure is likely deleterious, although not all deleterious variants disturb the protein structure.; (3) RP can precisely measure the impact of missense VUS variants on the protein secondary structure.; (4) MDS can measure the dynamic impact of missense VUS variants on the macroscopic properties of the protein structure. By using the protein structure as the only indication, RPMDS addresses the deleteriousness of missense VUS variants from an anchor very different from the principles used in existing in silico methods. The fact that RPMDS provides the lowest rate of deleterious prediction than these by other in silico methods indicates that RPMDS largely improves the over-prediction problem inherited in current in silico methods. Furthermore, the results from RPMDS are interpretable, as they are quantitative, visible, measurable, and reproducible, as reflected by their impact on the protein structure. With its high-throughput capacity, the use of the well-determined protein structure as the reference, and the use of known pathogenic and benign variants from the same gene as the training materials, RPMDS provides a promising new means for missense VUS variant classification, as demonstrated by our current study. While the definition of “deleterious” is more biological in nature and not equivalent to the more clinically oriented “pathogenic” definition, it does provide suggestive functional evidence for the potential pathogenic significance, as reflected by its disturbance of the protein structure. Integration of the evidence from other aspects will promote firm classification of missense VUS into either pathogenic or benign variants.

In summary, our study provides evidence for the deleteriousness of 193 *TP53* missense VUS based on their impact on TP53 structural stability.

## 4. Materials and Methods

### 4.1. Source of Missense VUS Variants

We used the ClinVar database as the source for *TP53* missense VUS variants because it provides VUS classification following ACMG/AMG guidelines [39]. The annotation information of genome position, base change, and coding change for each missense VUS was directly adapted from ClinVar (accessed on 9 April 2020).

### 4.2. Molecular Dynamics Simulation (MDS)

WT TP53 DBD crystal structure (PDB ID:2OCJ) was used as the template in the study [15]. The structure for each missense mutant TP53 was built using UCSF Chimera software and Modeller package ((University of California San Francisco, CA, USA) [42,43], following the procedures, in which the WT residue was substituted with the variant-changed residue in the template crystal structure and the variant residue orientation was chosen based on the highest probability/lowest energy. Each mutant TP53 DBD structure was simulated using GROMACS molecular dynamics software, version 2020 (University of Groningen, Netherlands) [44]. AMBER03 was chosen to model the protein complex and the zinc ion. The protein structure was situated in a 10 × 10 × 10 nm simulation box, solvated with SPC/E water, and neutralized with Cl^−^ ions. Each mutant structure consisting of SPC/E water and a single TP53 DBD contained ~99,000 atoms. The system was optimized with a steep descent algorithm for a 1 ns equilibration run at 298 K and 1 bar in the NPT ensemble using a Berendsen thermostat and barostat, and simulated for a 40 ns production run at 298 K and 1 bar in the NPT ensemble using a V-rescale thermostat and Parrinello-Rahman barostat [45]. The Verlet velocity algorithm was employed to integrate Newton’s equation of motion, with a time step of 2 fs. The Particle Mesh Ewald method was used to treat the long-range electrostatic interactions, with the cut-off distance set at 1.0 nm [46]. The LINC algorithm was applied to constrain the hydrogen bond at equilibrium lengths, and the trajectory frame of MD was saved every 30 ps [47].

### 4.3. Ramachandran Plot (RP)

The Ramachandran plot for each variant was generated from the MDS trajectory following the procedures [13]. Briefly, the torsional angle position by MDS was retained for each residue in the protein throughout the simulation. The Ramachandran plot was transformed into a density graph using Kernel density estimation, with a grid dimension of 32 × 32 to quantify the scatter plot. The average density of WT TP53 alleles, Benign (B) and Likely Benign (LB) variants was used as the training data, and standard deviation for each grid point was calculated. Missense VUS and Pathogenic (P) variants were compared to the training data and marked as significant “density deviation” if they were beyond the standard deviations.

## Figures and Tables

**Figure 1 ijms-22-11345-f001:**
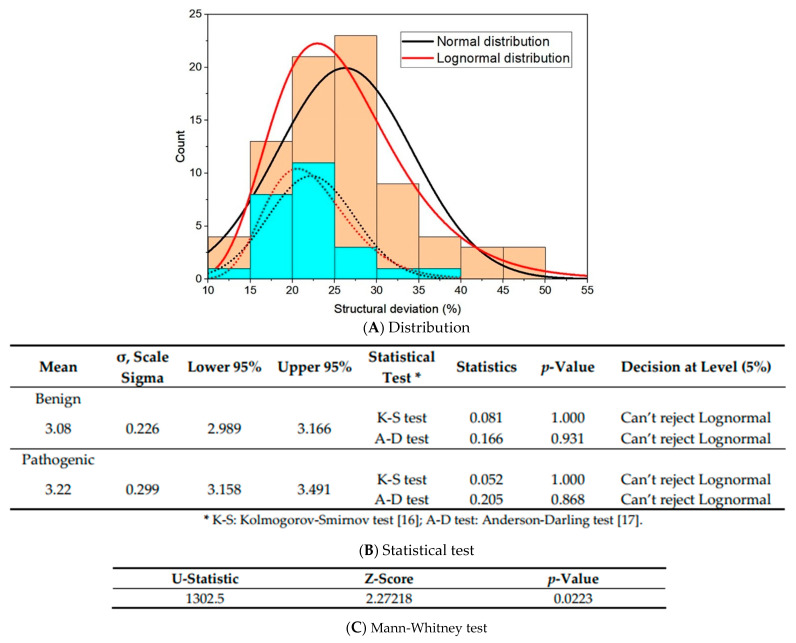
Goodness-of-fit test for 23 benign/likely benign and 80 pathogenic *TP53* variants. (**A**) Normal and lognormal distribution curves of the known benign/likely benign (cyan) and pathogenic variants (orange). The curve fits better in lognormal distribution (red) than in normal distribution (black) for both benign and pathogenic variants. (**B**) Statistics test. (**C**) Mann-Whitney test for comparing benign and pathogenic data.

**Figure 2 ijms-22-11345-f002:**
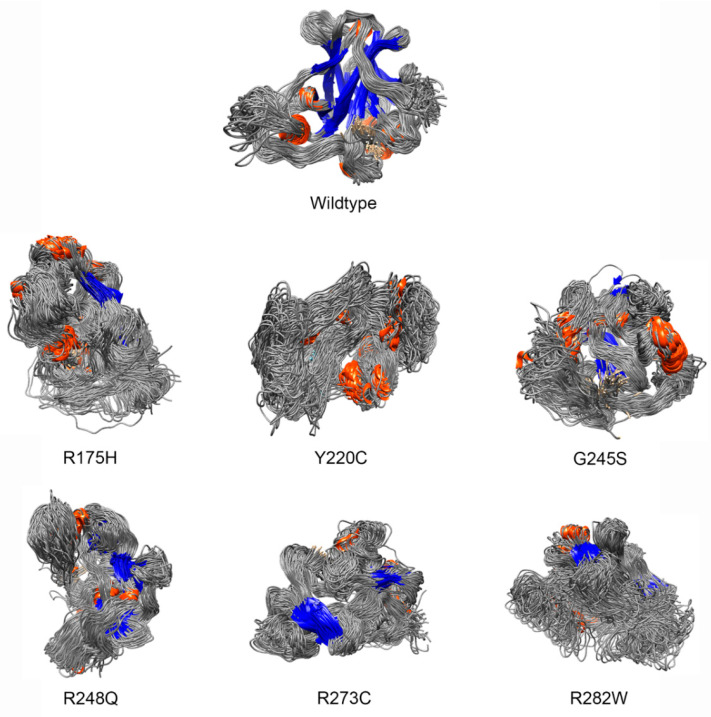
Examples of structural impact of known pathogenic variants classified by RPMDS. The known pathogenic variants of R175H, Y220C, G245S, R248Q, R273C, and R282W had a structural derivation of 43.7–49.9% from the WT TP53 structure, indicating their deleterious nature by destructing the TP53 structure. The structures of mutant TP53 and WT TP53 extracted from the last 10 ns of simulations were overlaid to reveal structural changes. Orange: α-helix; blue: β sheet; grey: the secondary structure linkers.

**Figure 3 ijms-22-11345-f003:**
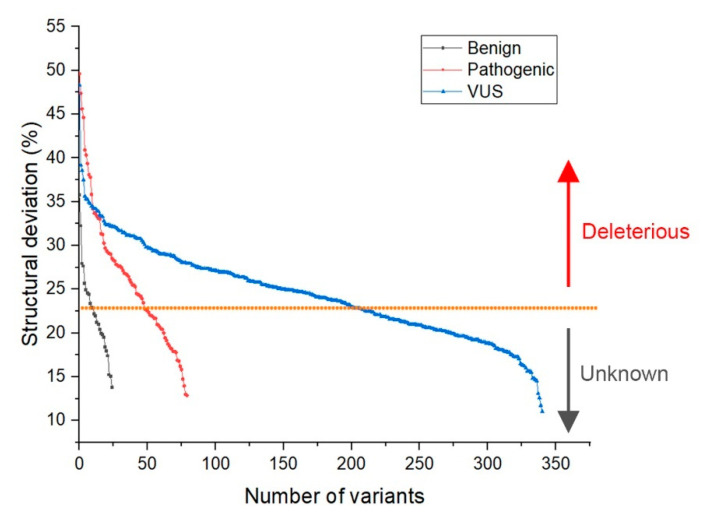
Deviation distribution between known benign, known deleterious pathogenic and classified deleterious missense VUS variants, showing the similar range of structural deviation between the two types of deleterious variants. The variants above and below the orange line were classified as “deleterious” and “unknown”, respectively. Black: known benign; Red: known deleterious pathogenic; Blue: classified deleterious missense VUS.

**Figure 4 ijms-22-11345-f004:**
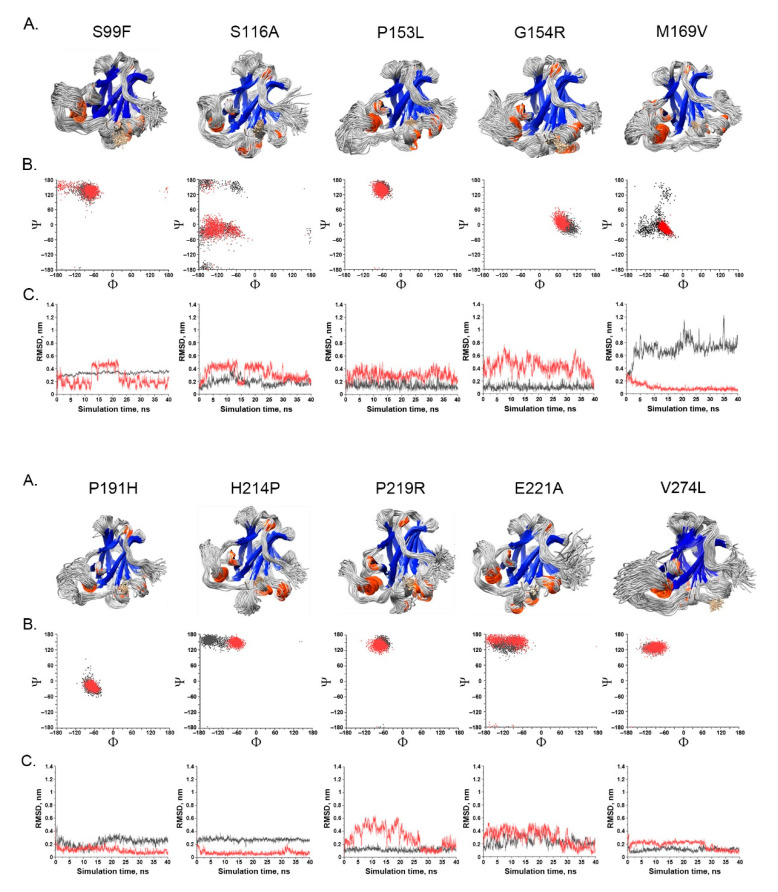
Spatial change of the substituted residues by missense VUS variants. (**A**) Superimposed variant protein structure; (**B**) Ramachandran scatter plot of the WT residue (black) and the substituted residue (red); (**C**) RMSD plot of the substituted residue (red) relative to the global protein structure. The substituted residues (red) fluctuated at different positions in comparison to the WT residues (black), implying the substituted residues caused different spatial coordination. See detailed description in Results.

**Figure 5 ijms-22-11345-f005:**
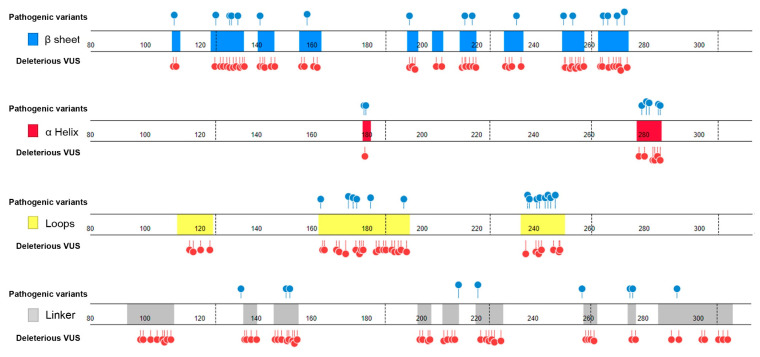
Distribution of deleterious VUS variants along TP53 secondary structure. The deleterious missense VUS variants were distributed across all β sheets, loop 1 and loop 2, and all linkers where a few known pathogenic variants were present. blue: β sheet; red: α helix; yellow: loops; grey: linkers; lollipop in blue: known pathogenic variants; lollipop in red: deleterious missense VUS variants.

**Figure 6 ijms-22-11345-f006:**
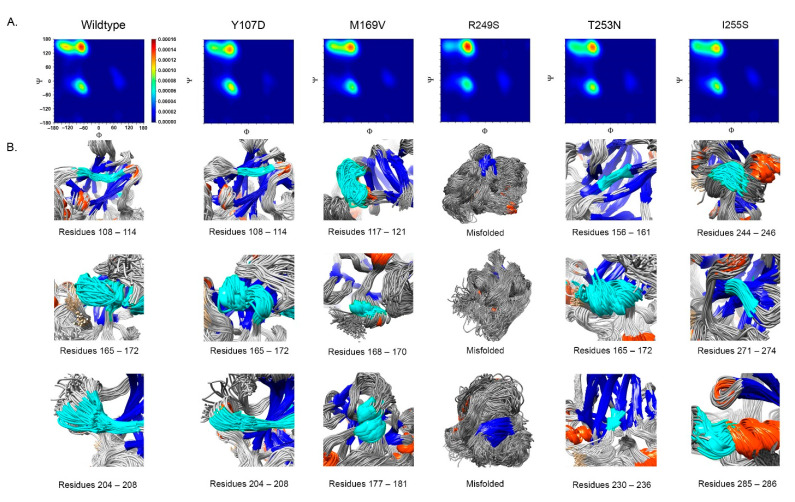
Impact of missense VUS on TP53 local structure. (**A**) Ramachandran density plot for missense VUS M169V, N239T, R249S, I255S, and P278R. The β strand regions (ϕ, ψ = (−130, 140)) for Y107D, M169V, R249S, T253N, and I255S were notably diminished, and the PII-spirals (ϕ, ψ = (−45, +135)) for R249S were intensified in comparison to the wildtype. The color change from red to blue represented the density from high to low, respectively. (**B**) Graphical illustration for local structural changes in Y107D, M169V, R249S, T253N, and I225S. WT residue positions were shown only for Y107D with the residues 108–114, 165–172, and 204–208. R249S showed global misfolded structure of TP53. Blue: β strand; orange: α helix; grey: linker joint; cyan: additional structural features.

**Table 1 ijms-22-11345-t001:** Comparison of RPMDS-based missense VUS classification with other methods.

**(A) Comparison with Updated ClinVar Classification**				
**Variant**		**ClinVar Classification**		**RPMDS Classification**
**Nucleotide**	**Amino Acid**	**Original**	**New**	
c.706T>G	Y236D	VUS	Pathogenic	Deleterious
c.695T>A	I232N	VUS	Pathogenic	Deleterious
c.794T>A	L265Q	VUS	Pathogenic	Deleterious
c.413C>T	A138V	VUS	Conflict	Deleterious
c.422G>T	C141F	VUS	Conflict	Deleterious
c.434T>C	L145P	VUS	Conflict	Deleterious
c.526T>A	C176S	VUS	Conflict	Deleterious
c.556G>A	D186N	VUS	Conflict	Deleterious
c.581T>G	L194R	VUS	Conflict	Deleterious
c.626G>A	R209K	VUS	Conflict	Deleterious
c.814G>A	V272M	VUS	Conflict	Deleterious
c.931A>C	N311H	VUS	Conflict	Deleterious
c.431A>T	Q144L	VUS	Conflict	Undefined
c.452C>G	P151R	VUS	Conflict	Undefined
c.658T>C	Y220H	VUS	Conflict	Undefined
c.730G>T	G244C	VUS	Conflict	Undefined
c.928A>G	N310D	VUS	Conflict	Undefined
**(B) Comparison with Different In Silico Methods in Classifying 340 Missense VUS**				
**Methods**	**Classification**			
	**Deleterious ***	**Rate (%)**	**Undefined ****	**Rate (%)**
RPMDS	193	56.8	147	43.2
Polyphen2_HVAR_pred	202	59.4	138	40.6
Polyphen2_HDIV_pred	217	62.9	123	37.1
PROVEAN_pred	233	68.5	107	31.5
LRT_pred	242	71.2	98	28.8
SIFT_pred	268	78.8	72	21.2
MutationTaster_pred	287	84.4	53	15.6
M-CAP_pred	335	98.5	5	1.5
FATHMM_pred	340	100	0	0
MetaSVM_pred	340	100	0	0
MetaLR_pred	340	100	0	0

* These are named as Deleterious: Polyphen2_HDIV/Polyphen2_HVAR: Probably damaging and Possibly damaging; MutationTaster: Disease causing. ** These are named as Undefined: SIFT/FATHMM/MetaSVM/MetaLR/M-CAP: Tolerate; Polyphen2_HDIV/Polyphen2_HVAR: Benign; LRT/PROVEAN/FATHMM: Neutral; MutationTaster: Polymorphism.

## Data Availability

Data generated from the study are provided as Appendix A.

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
