# Peer review of "Comprehensive Identification of Deleterious TP53 Missense VUS Variants Based on Their Impact on TP53 Structural Stability"

_ijms, 2021, doi:10.3390/ijms222111345_

Round 1

Reviewer 1 Report

The manuscript presented here aims to propose a tool aimed at predicting the pathogenicity of VUS mutations involving a TP53 protein domain. In my opinion, the study suffers from a fundamental error that undermines its scientific validity. In fact, to define a cutoff through which to discriminate the potential pathogenicity of a VUS mutation, the authors write the following: "the density deviation of the 80 known pathogenic variants were plotted against the normal and lognormal distribution curves for Anderson-Darling and Kolmogorov-Smirnov goodness of fit tests [...] Here, the "structural deviation" was defined as the Ramachandran plot difference between pathogenic and the trained data, of which the trained data contained the average Benign and wild type (WT) Ramachandran density plots. A significant deviation with a mean of 3.370 was obtained, with a sigma scale". The problem is that the so-called "trained data" dataset consists of only 8 variants classified by Clinvar as benign + the set of wild type structures of pathogenic variants. This approach generates a bias, since it is taken for granted that any allelic variant causes a "structural deviation" from the wild type allele. The authors should compare the results of their predictive test comparing pathogenic versus missense variants not associated with cancer risk, and thus theoretically benign, by searching more of the 8 benign variants used here in other public databases on human genetic variability.

Author Response

Reviewer 1

The manuscript presented here aims to propose a tool aimed at predicting the pathogenicity of VUS mutations involving a TP53 protein domain. In my opinion, the study suffers from a fundamental error that undermines its scientific validity. In fact, to define a cutoff through which to discriminate the potential pathogenicity of a VUS mutation, the authors write the following: "the density deviation of the 80 known pathogenic variants were plotted against the normal and lognormal distribution curves for Anderson-Darling and Kolmogorov-Smirnov goodness of fit tests [...] Here, the "structural deviation" was defined as the Ramachandran plot difference between pathogenic and the trained data, of which the trained data contained the average Benign and wild type (WT) Ramachandran density plots. A significant deviation with a mean of 3.370 was obtained, with a sigma scale". The problem is that the so-called "trained data" dataset consists of only 8 variants classified by Clinvar as benign + the set of wild type structures of pathogenic variants. This approach generates a bias, since it is taken for granted that any allelic variant causes a "structural deviation" from the wild type allele. The authors should compare the results of their predictive test comparing pathogenic versus missense variants not associated with cancer risk, and thus theoretically benign, by searching more of the 8 benign variants used here in other public databases on human genetic variability.

Answer:

We thank reviewer for this valuable comment. Here, from the ClinVar recent updated list (accessed 23/09/2021), we identified additional 15 new Benign/likely Benign variants not included in the previous version used in our study, of which 13 were reclassified from the previous VUS. To avoid uncertainty of the data used in our analysis, we only used the variants classified as benign, likely benign, VUS, likely pathogenic and pathogenic following the ACMG guidelines by ClinVar. The uncertain variants classified by other databases such as IARC were based on different in silico prediction methods, which often over-predict VUS as deleterious as shown in our analysis (Table 2B, Supplementary table 4). Using these VUS data as input as training data for RPMDS could disturb the quality of our system. Therefore, we are reluctant to use the classification from other in silico methods as input trained data for our RPMDS pipeline.

In the revision, we had 23 Benign variants from the original 8, 80 Pathogenic variants and 340 VUS. With the inclusion of new benign variants in the trained data, the deleterious variants changed accordingly as predicted by the reviewer with potentially better quality of the results. We revised the data in related tables, figures and supplementary tables accordingly as marked in the revision. In total, 193 out of 340 VUS were classified as deleterious VUS, comparing to the original 204 out of 353.

Reviewer 2 Report

Tam et al. employ their recently developed Ramachandran Plot-Molecular Dynamics Simulation (RPMDS) to predict the effect of missense mutations in the TP53 gene on the structural stability of the encoded tumor suppressor protein p53. The study rests on the rational that lowered structural stability correlates strongly with loss of p53 tumor suppressor function. The authors determine thresholds for their predictive model using publicly available structural data from wild-type p53 and missense mutations for which a loss of function is well-established. For 353 missense mutations classified as variants of unknown significance (VUS) in the ClinVar database, the authors use RPMDS to predict the mutational impact on p53 structure as a surrogate for protein function. The authors predict for 204 out of 353 VUS a substantial reduction in protein stability and, thus, loss of function. Further, the authors attempt to validate their findings by updated ClinVar database entries that became available within a year after the authors started their analyses.

Overall, the authors’ approach is intriguing and the data could complement the current ClinVar database by providing predictions for functional consequences of TP53 mutations for which current data is sparse. However, I felt that further validation data is required before publication.

Major:

  • The only major shortcoming of the current study is the limited validation performed by the authors.

First, Figures 1 and 3 should be complemented with data on the known benign variants and Figure 3 should further be complemented by data on the predicted benign VUS.

Second, the authors should extend their validation data, which could be done using wet-lab experiments or by using publicly available data. The latter includes TCGA data which contains RNA-seq data for tumor samples that harbor at least some of the 204 variants identified as loss of function by the authors. The authors should show for at least some of those variants that p53 target gene expression is lower in samples harboring such mutations when compared to p53 wild-type and predicted benign VUS. Known loss of function and known benign mutations can serve as controls.

Minor

  • Given that the authors rely on simulations/predictions, they should tone down phrases like “our study provides evidence” to “our data suggest” throughout the manuscript.

Author Response

Reviewer 2

Tam et al. employ their recently developed Ramachandran Plot-Molecular Dynamics Simulation (RPMDS) to predict the effect of missense mutations in the TP53 gene on the structural stability of the encoded tumor suppressor protein p53. The study rests on the rational that lowered structural stability correlates strongly with loss of p53 tumor suppressor function. The authors determine thresholds for their predictive model using publicly available structural data from wild-type p53 and missense mutations for which a loss of function is well-established. For 353 missense mutations classified as variants of unknown significance (VUS) in the ClinVar database, the authors use RPMDS to predict the mutational impact on p53 structure as a surrogate for protein function. The authors predict for 204 out of 353 VUS a substantial reduction in protein stability and, thus, loss of function. Further, the authors attempt to validate their findings by updated ClinVar database entries that became available within a year after the authors started their analyses.

Overall, the authors’ approach is intriguing and the data could complement the current ClinVar database by providing predictions for functional consequences of TP53 mutations for which current data is sparse. However, I felt that further validation data is required before publication.

Major:

  • The only major shortcoming of the current study is the limited validation performed by the authors.

First, Figures 1 and 3 should be complemented with data on the known benign variants and Figure 3 should further be complemented by data on the predicted benign VUS.

Answer:

We thank reviewer’s comments. We have updated the Figure 1 and Figure 3 as recommended,  with Benign and Unknown included in Figure 3. As indicated in our previous publication (PMID: 33363700) that not all pathogenic variants cause significant structural change, therefore, there can be overlapping of structure deviation between pathogenic and benign variants. Thus, in Figure 3, we named these below the cut-off line as “Unknown VUS” instead of “predicted benign VUS”.  In reflecting the increased benign variants in the trained data, the order of certain deleterious VUS were slightly changed from the previous submission accordingly (Supplementary table 3).

Second:

  • the authors should extend their validation data, which could be done using wet-lab experiments or by using publicly available data. The latter includes TCGA data which contains RNA-seq data for tumor samples that harbor at least some of the 204 variants identified as loss of function by the authors. The authors should show for at least some of those variants that p53 target gene expression is lower in samples harboring such mutations when compared to p53 wild-type and predicted benign VUS. Known loss of function and known benign mutations can serve as controls.

Answer:

We thank reviewer’s valuable comments. However, for the following reasons, we are reluctant to perform the analysis:

  1. It needs to unearth the entire TCGA data in order to identify these with TP53 VUSs included in our VUS list. This will require substantial expertise and computational power. We estimate at least 2 months will be required to complete this work. If lucky, we may identify some targeted VUS there. However, there is no guarantee to obtain the expected results. As such, we may end up with no targeted VUS identified;
  2. As transcriptional factor, TP53 regulates multi-thousands of gene’ expression and mutated TP53 can certainly cause altered expression of these TP53 regulated genes. By following the altered expressed genes, we can identify these as the evidence to approve or disapprove the predicted deleterious VUS in TP53. However, the question is how to be sure that the genes with altered expression is caused with the deleterious VUS in TP53. It has been questioned for the highly conserved TP53 regulated genes: a comprehensive comparison of the TP53 regulated genes reported by major studies suggests that the so-called “TP53 regulated genes” could be highly unreliable [Fischer M. Census and evaluation of p53 target genes. Oncogene. 2017 13;36(28):3943-3956]. Therefore, even altered expression of the proposed TP53-regulated genes could be identified from the RNAseq data containing VUS-haboringTP53, the connection between the change expression and the VUS-haboringTP53 could not be certain.

In saying so, we plan to test the concept in our follow-up study. If this could be the case, it will certainly provide a powerful tool for validating the predicted VUS.

Minor

  • Given that the authors rely on simulations/predictions, they should tone down phrases like “our study provides evidence” to “our data suggest” throughout the manuscript.

Answer:

Fully agree. We made the change across the revision accordingly as recommended.

Reviewer 3 Report

Comprehensive identification of deleterious TP53 missense VUS variants based on their impact on TP53 structural stability

In this article, the authors have used combination of Ramachandran Plot-Molecular Dynamics Simulation (RPMDS) to predict how variants of unknown significance (VUS) alters the structural integrity of p53 protein. Additionally, the authors compared their methods with other available in-silico methods for similar predictions. The authors have done commendable job to show that simulation can predict the effect of mutation on local environment around it or on overall global structure of the protein. I would recommend acceptance of this article in its current form.

Minor Point:

Please fix a typo on line 52, it should be wild-type and not wide-type.

Author Response

Reviewer 3

In this article, the authors have used combination of Ramachandran Plot-Molecular Dynamics Simulation (RPMDS) to predict how variants of unknown significance (VUS) alters the structural integrity of p53 protein. Additionally, the authors compared their methods with other available in-silico methods for similar predictions. The authors have done commendable job to show that simulation can predict the effect of mutation on local environment around it or on overall global structure of the protein. I would recommend acceptance of this article in its current form.

Answer:

We appreciate the author for reviewing this paper.

Minor Point:

Please fix a typo on line 52, it should be wild-type and not wide-type.

Answer:

We have changed the mistake as the reviewer indicated.

Round 2

Reviewer 1 Report

The authors have tried to improve the work and their efforts are appreciated. However, there are a number of important problems that may detract from the scope of their analysis. In fact, the authors should show with a statistical test if the distribution of their score is SIGNIFICANTLY different between benign variants and pathogenic variants (and not the "wild-type"... as I said, "wild-type variants" do not exist: at most they should talk about "wild-type alleles"). Also, they should show what is the power and effectiveness of the test to discriminate between a known pathogenic variant and a benign one with a validation dataset.

Author Response

Question

The authors have tried to improve the work and their efforts are appreciated.

Answer

We thank the reviewer’s encouragement. The comments made by the reviewers promoted us to try our best to improve the quality of our study.

Question

The authors have tried to improve the work and their efforts are appreciated. However, there are a number of important problems that may detract from the scope of their analysis. In fact, the authors should show with a statistical test if the distribution of their score is SIGNIFICANTLY different between benign variants and pathogenic variants

Answer

We thank the reviewer for suggesting statistical test for the benign and pathogenic variants data. Here, we have used non-parametric Mann-Whitney test and showed that the two sets of data is significantly different [Z score of 2.27 and p-value of 0.0223 (Figure 1C)].

Question

and not the "wild-type"... as I said, "wild-type variants" do not exist: at most they should talk about "wild-type alleles".

Answer

We thank the reviewer for the comment. In the revision, we have changed all “wild type variants” to “wild type alleles”.

Question

they should show what is the power and effectiveness of the test to discriminate between a known pathogenic variant and a benign one with a validation dataset.

Answer

We thank the reviewer’s comment. However, as indicated in our first revision, we have exhausted all known pathogenic variants and benign variants in ClinVar database for training our model. With the limited numbers, it is difficult to analyze the power of discriminating between pathogenic and benign variants. However, we identified 3 VUS used in our analysis but reclassified as pathogenic variants, and 9 VUS reclassified as conflict in the latest updates in ClinVar database (Table 2A). We were unable to locate more new benigns as we used all benign/likely benign variants as training data to strengthen benign determination.